# What Is the Inpatient Cost of Hip Replacement? A Time-Driven Activity Based Costing Pilot Study in an Italian Public Hospital

**DOI:** 10.3390/jcm11236928

**Published:** 2022-11-24

**Authors:** Andrea Fidanza, Irene Schettini, Gabriele Palozzi, Vasileios Mitrousias, Giandomenico Logroscino, Emilio Romanini, Vittorio Calvisi

**Affiliations:** 1Mininvasive Orthopaedic Surgery, Department Life, Health and Environmental Sciences, University of L’Aquila, 67100 L’Aquila, Italy; 2Department of Management and Law, Tor Vergata University of Rome, 00133 Roma, Italy; 3Faculty of Medicine, School of Health Sciences, University of Thessaly, 41500 Larissa, Greece; 4RomaPro Center for Hip and Knee Arthroplasty, Polo Sanitario San Feliciano, 00166 Rome, Italy

**Keywords:** TDABC, ABC, value-based health care, VBHC, hospital costs, costs and costs analysis, THA, outcomes, decreasing costs, optimization

## Abstract

The emphasis on value-based payment models for primary total hip replacement (THA) results in a greater need for orthopaedic surgeons and hospitals to better understand actual costs and resource use. Time-Driven Activity-Based Costing (TDABC) is an innovative approach to measure expenses more accurately and address cost challenges. It estimates the quantity of time and the cost per unit of time of each resource (e.g., equipment and personnel) used across an episode of care. Our goal is to understand the true cost of a THA using the TDABC in an Italian public hospital and to comprehend how the adoption of this method might enhance the process of providing healthcare from an organizational and financial standpoint. During 2019, the main activities required for total hip replacement surgery, the operators involved, and the intraoperative consumables were identified. A process map was produced to identify the patient’s concrete path during hospitalization and the length of stay was also recorded. The total inpatient cost of THA, net of all indirect costs normally included in a DRG-based reimbursement, was about EUR 6000. The observation of a total of 90 patients identified 2 main expense items: the prosthetic device alone represents 50.4% of the total cost, followed by the hospitalization, which constitutes 41.5%. TDABC has proven to be a precise method for determining the cost of the healthcare delivery process for THA, considering facilities, equipment, and staff employed. The process map made it possible to identify waste and redundancies. Surgeons should be aware that the choice of prosthetic device and that a lack of pre-planning for discharge can exponentially alter the hospital expenditure for a patient undergoing primary THA.

## 1. Introduction

Osteoarthritis (OA) is one of the most common chronic degenerative diseases affecting a wide range of the population, as well as one of the most frequent causes of disability in the elderly. Symptomatic osteoarthritis is estimated to affect, in Italy alone, at least 4 million people, with a public annual cost of approximately EUR 6.5 billion [1,2]. Total hip arthroplasty (THA) is considered the best solution for the treatment of patients with severe hip osteoarthritis [3].

In 2019, 118,673 hip replacement surgeries were performed in Italy, and this number is increasing at a rate of approximately 2.7% per year: in 2017 and 2018, the number of surgeries were 112,375 and 115,308, respectively [4,5]. This pattern is in line with the Organization for Economic Co-operation and Development (OECD) data, which reveals a significant rise in THAs for the majority of OECD nations [6]. In the pre-COVID period, the reimbursement by the Italian National Health System related to arthroplasty surgery was about EUR 1,625,853,413 [7]. Due to its significant effect on healthcare system costs and the high frequency and demand for this surgical procedure, many authors have defined THA complications as a real health emergency [8,9]. This is also the reason for an increasing interest in understanding the real cost related to this procedure.

In fact, using resources for acquiring important information for the entire activity of healthcare organizations is crucial in the era of value-based healthcare. Tools for management accounting might be regarded helpful for information gathering in the context of healthcare, in order to accomplish this purpose. The costs of medical treatments calculated using Activity Based Costing (ABC) tend to have more accuracy in the computation of resource consumption than standard cost accounting systems, among other techniques of cost calculation or reimbursement, such as Diagnosis-Related Groups (DRG) [10]. Furthermore, the capacity to precisely identify expenses at the level of the treatment process and manage the complexities associated with accounting in the healthcare sector make the Time-Driven Activity-Based Costing (TDABC) [11,12] the most effective and straightforward instrument, even when compared to the traditional ABC. TDABC is an innovative approach used to measure costs more accurately by estimating the amount of time and cost per unit of time each provider uses during a care episode. For example, if a staff member spends 30 min with a patient and that staff member’s time cost is EUR 100 per hour, the cost of interacting with that patient is EUR 50.

In addition, the use of the TDABC method estimates the practical capacity (i.e., actual production time) of each element providing capacity (operator and equipment) and the average time required for each element to carry out the action, on the basis of observation, data collection, and questionnaires. The improved process consistency makes TDABC particularly appropriate for the surgical area [13].

This “bottom-up” accounting method makes it possible to identify transparent analyses of the entire care cycle by adding up the individual costs of all the resources used by a single patient [13].

Furthermore, this model allows to redesign the process in order to reduce costs, incorporate new activities in the care cycle, make changes and, above all, it allows to compare the best pathway and offer the patient the best available solution, identifying areas for improvement in terms of time, goods consumed, and the activities carried out.

Clinical management, including medical professionals and support employees, would greatly benefit from this development in healthcare since it would make it easier for them to quickly assess a treatment’s efficacy and resources usage. TDABC’s process mapping might reveal which processes offer the most value, if waste can be reduced, and whether resources are being underutilized from the standpoint of redesigning the delivered healthcare process to create high value for the patient. According to numerous authors [13,14,15,16], its use in orthopaedic surgery is very suited. The ability of an activity-based methodology to not only supply more information but also to provide better detail and higher timeliness of the same, constituting a legitimate support to the decision-making process, is another point on which all of these authors agree. Furthermore, the use of this technique lowers the percentage of unspecified allocated overhead costs, and process mapping makes it easier to oversee every step of the process efficiently, taking corrective action as needed [17]. The resources used and the actions performed are precisely specified, ensuring that the cost analysis is accurate and complete [18].

A recent systematic review concluded that TDABC can help overcome a key challenge associated with current cost accounting methods and should be gradually incorporated into functional systems [12].

Additionally, the information gathered and the methods used to identify expenses improve the transparency in the business management that enable, as shown by the research by Demeere N. et al. [19], an internal examination aimed at establishing a reference benchmark and creating value.

The goal of this pilot study is to understand the true cost of a total hip replacement using the TDABC in an Italian public hospital and to comprehend how the adoption of this method might enhance the process of providing healthcare from an organizational and financial standpoint.

## 2. Materials and Methods

### 2.1. Study Design

During 2019 a prospective experimental case study [20,21] was conducted in a public hospital in central Italy. The orthopaedic department under observation is made up of 26 hospital beds, 7 orthopaedic and trauma surgeons, and 4 residents; it is part of a regional HUB and performs about 1600 orthopaedic surgeries per year.

Using the TDABC, details regarding all the activities, consumables, and participating healthcare professionals were gathered. Seven steps have been methodologically introduced, as is specifically mandatory for the use of this tool [12], and they are identified by the increasing number in brackets of the following paragraphs.

All procedures were performed in accordance with ethical standards, and the study protocol was approved by the Internal Review Board of authors’ affiliated Institution (authorization number 22/2022).

### 2.2. Study Participants

(1) Inclusion criteria: patients 60–80 years old, suffering from primary hip arthritis with indication for THA. Patients with concomitant femoral neck fractures, cemented prostheses, intraoperative fractures, or systemic complications (such as cardiopulmonary diseases, which would have increased standard surgical times) were excluded.

### 2.3. Measurement

To comprehend how patients move through the care cycle and quantify the usage of human resources by activity, a process map for primary hip replacement was created from the entrance in the operating room to the exit (2).

Direct observation, interviews, and multidisciplinary care plan validation sessions with frontline personnel were used to build process maps with time estimates for each stage. The necessary resources for each process step (such as staff and consumables like implants) were noted (3).

We calculated the overall expenses over a patient’s cycle of care after estimating the cost of providing each service based on the time needed for each resource type (4).

The questions performed to the healthcare staff are listed in the Appendix A (Table A1).

### 2.4. Costs Analysis of Hip Arthroplasty

(5) The average price stated by the regional fee schedules as compensation for services rendered in pre-hospitalization was used as a reference point to determine the cost of pre-hospitalization exams [22]. The cost of the majority of consumer goods used throughout the entire therapeutic process was provided by the Director of hospital’s pharmacy by filling out a pre-set table based on the information needs derived from the process maps; the cost of the prosthetic device was obtained by extracting the price from the purchase of regional tender. The hourly cost of an active operating room used for major hip surgery, net of material, and labour costs, was derived from the literature [23], as well as the average cost of a day of hospitalization [24]. (6) Comparing the average monthly pay of the operators with the actual amount of time spent delivering the health service, the capacity cost rate [11], defined as practical capacity of each active operator, was determined. This method was used for all the operators present, taking into account the different remuneration (7). It is important to point out that in the Italian system, the salary for clinical staff in a public hospital is regulated nationally on a monthly basis. There is no difference pay based on the procedures carried out; rather, it is dependent on the total number of hours worked each month and the operators’ seniority.

Finally, to define the total cost of hip arthroplasty, from the admission to the patient’s discharge, all the calculated expenses (pre-operative tests + hospitalization + theatre + general consumer goods + prosthesis + staff employed) were added up.

## 3. Results

Ninety patients who met our inclusion criteria were included in this study. The cementless prosthetic implant was the same for each patient and the surgical team was the same for all operations. The average of actions and time spent on these patients made up the process map shown in Figure 1: this diagram outlines the arthroplasty operating day including anaesthesia preparation, surgical preparation, and surgery. From the moment the patient enters the pre-operative room until the last radiographic control following the surgery, the estimated time for the intervention is, on average, 90 min.

The healthcare delivery process of THA in the hospital analysed involved seven healthcare professions, in particular:-Three orthopaedic surgeons (one involved for 55 min and two involved for 39 min);-One anaesthesiologist (involved all the time);-One nurse dedicated to the anaesthesiologist (involved all the time);-One surgical nurse (involved 85 min);-One general nurse (involved 85 min).

The pre-operative tests, including blood tests, chest X-ray, pelvic X-ray, ECG, and anaesthesia evaluation reached a total cost of EUR 90.29 per patient, according to the regional fee.

The total cost related to the personnel involved in the THA implantation procedure is EUR 201.34 and it is shown in detail in Table 1. Considering the average of the wages specified in the Italian National Labour Contract for operators with that level of experience, 36 h per week were calculated for nurses, and 38 h per week for doctors.

The cost of the consumables charged for each operation performed by healthcare professionals was grouped by stage, and it represents an expense of EUR 97.02 (Table 2).

The cost of each consumable item is listed in the Appendix A (Table A2).

The final cost of the implanted prosthetic device was EUR 3029.208 according to the regional tender, and it is analysed in Table 3.

The cost of an active operating room used for major hip surgeries, minus material and labour expenses, was calculated by Cinquini et al. [23], who estimated an hourly cost of EUR 90 for the theatre. By multiplying the hourly cost by the amount of time spent in the operating room for the procedure (90 min), a cost of EUR 135 was obtained.

The average inpatient stay was 3.7 days. On 2007, the Italian Ministry of Economy and Finance estimated the average cost of hospitalization to be EUR 674 per day [24]. The final cost of hospitalization was calculated by multiplying the average daily cost of a day by the number of days spent in the hospital by the patient, resulting in EUR 2493.80.

The total cost of THA from pre-operative tests to discharge is EUR 6002.06. It is presented in detail in Table 4.

## 4. Discussion

The world health system is facing an unprecedented period of change and crisis. The ongoing war in Eastern Europe and the post-COVID crisis have also caused an increase in national healthcare spending, predictably far greater than inflation. This unsustainable health care expenditure has increased the demand for providing high quality care while reducing the costs of delivering these outcomes. The main finding of this pilot study is that the TDABC methodology can also be applied in an Italian public hospital and provides a complete and detailed description of the patient’s path, of the staff involved, and an accurate understanding of the operation costs.

The value of health care, defined by health outcomes achieved for every dollar spent [25], can be improved if costs and outcomes are measured in sufficient detail to assess the impact of changes in care systems and processes [26,27]. It is not just economic stuff, since knowing organizational and clinical details can allow healthcare professionals to redesign care processes with a patient-centred view, thus providing the best possible care using resources efficiently.

An accurate understanding of costs is important for the effective implementation of cost saving strategies. Elective orthopaedic surgeries are often standardized and in order to increase the efficiency of such surgical procedures it is essential to develop process maps for each step of care in the total joint replacement. In this way, we will be able to identify redundancies and welfare inefficiencies, whose financial impact was not previously detectable due to the lack of detailed analysis of the processes obtained with traditional accounting methods. On the other hand, the times and consumables used are certainly different, depending on the surgeon’s background, on the adhesion of the operating team to the most recent evidence-based scientific literature, or to guidelines. It is crucial to point out that regardless of the structure or volume of surgeons, standardization is associated with better processes and outcomes for patients undergoing THA and thus process mapping can also help improve procedures, increase productivity, and raise the number of hip replacement surgeries performed. The mapping of the clinical path, in fact, allows us to understand the connections between the activities, operators, roles, and responsibilities of the care delivery cycle. In addition, this allows service providers to have a better awareness of the costs related to certain services and allows them to evaluate the effects of changes to support systems and procedures.

The TDABC has been described by a number of authors as a managerial tool that promotes collaboration between medical professionals and support staff by outlining every step of the value-creation process [17,18,19]. In our experience, the total composition of the cost of a hospitalized patient to undergo THA has two major economic items: the component that has the greatest impact on this value is the cost of the prosthetic device, which alone represents 50.4% (about EUR 3000) of the total cost, followed by costs relating to hospitalization, which constitute 41.5% (about EUR 2500).

Our results agree with the literature showing that the largest and most common direct cost is the purchase price of the implant [28]. This is a common finding both in the literature on hip and knee arthroplasty [29], and in shoulder prostheses [30]. Robinson et al. showed that the average cost of the implant per case can range from USD 2392 to USD 12,651 for total hip replacement procedures [31].

Haas et al. [32] recently noted that hospitals using a joint committee of hospital administrators and surgeons to negotiate prices with vendors paid 17% less for implants than institutions without a joint purchasing committee.

Therefore, with a view to improving the efficiency or comparing costs with other institutes and methodologies, these data highlight the importance of rationalizing purchases and, where possible, reducing hospital stays.

While directly decreasing medication and personnel costs may not be feasible, indirectly lowering costs by reducing length of stay must be an area for improvement. It is proven that the common reasons why patients need to stay an extra day in hospital after joint replacement are outdated customs, unscientific fears and, at best, problems related to pain and social support, all potentially responsive to initiatives of quality improvement [33]. Strategies to reduce length of stay after THA have to start the first moment surgery is considered, ensuring adequate postoperative social support and setting realistic expectations about pain management [30].

Finally, to achieve higher-value care for patients undergoing THA, TDABC costs must be linked to patient experience, quality of life, and functional outcomes, which is the object of our future research. Nevertheless, this study is an important step toward introducing clarity into the cost conundrum of THA and will hopefully stimulate further research into this increasingly important topic.

Obviously, this study has limitations. First of all, as a pilot experiment it is a small-scale preliminary observational study undertaken to decide how and whether to start a large-scale project, which could aim at collecting data from multiple institutes in order to integrate and compare the results obtained in a public hospital with those obtainable in a private or university hospital. Secondly, it is important to underline that TDABC does not contain all expenses calculations usually included in a DRG-based reimbursement: it is a method to determine the cost of the process. The object cost is the healthcare delivery process of total hip replacement in the hospital. From this perspective, it may appear that the TDABC process does not fully account for all indirect and THA-related costs (such as administrative, research, or sterile treatment), nor time spent caring for the patient outside the hospital, such as medication, rehabilitation, and social support, but it does account for facilities, equipment, information technology, and most other traditional “overhead” costs [34]. In addition, our exclusion criteria reflect the assumption that surgery and postoperative recovery are without complications, which obviously would add costs to both staff and medications, blood transfusions, any new surgeries, and, in any case, a longer length of stay, which would certainly amplify the costs.

## 5. Conclusions

Time-Driven Activity-Based Costing provides an accurate assessment of costs in the field of hip replacement. The implementation of this methodology gives us more precise and detailed expense advice, a clear path of the patient, and information on the times and staff involved in each activity.

A careful choice of the prosthetic device among the wide range of offers on the market and a premeditated planning of the discharge can drastically reduce the hospital costs of a THA, without cutting staff or medications.

## Figures and Tables

**Figure 1 jcm-11-06928-f001:**
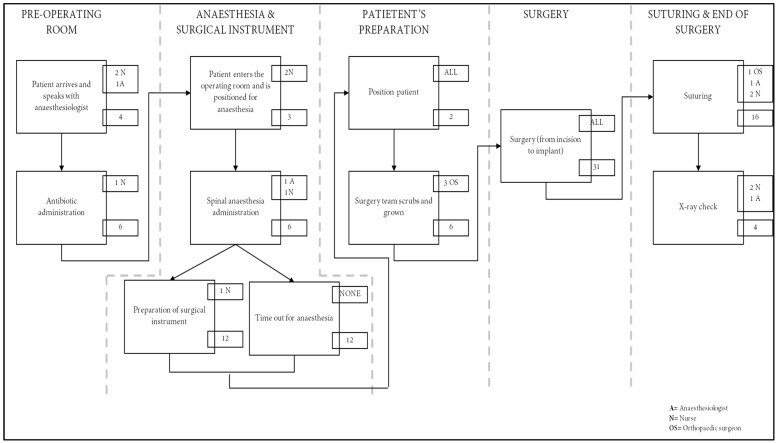
Process map of THA. The large boxes represent activities with arrows indicating sequence. The personnel ID is in the upper smaller boxes (see legend) while the numbers in the smaller boxes correspond to minutes used per activity.

**Table 1 jcm-11-06928-t001:** The cost refers to the total minutes spent by each health worker during the procedure.

Helthcare Professional	Monthly Pay	Total Time Spent	Total Cost
Orthopaedic Surgeons (x 2)	EUR 5330	39 min for each surgeon	EUR 45.59
Orthopaedic surgeon (x 1)	EUR 5330	55 min	EUR 32.14
Anaesthesiologist	EUR 6000	90 min	EUR 59.21
Nurses (x 2)	EUR 2052	85 min for each nurse	EUR 40.80
Nurse (x 1)	EUR 2052	90 min	EUR 21.60
TOTAL COST			EUR 201.74

**Table 2 jcm-11-06928-t002:** The grouped-by-stage cost of consumables charged for each operation performed by healthcare professionals.

Stage	Cost of Consumables
Pre-operating room	EUR 8.44
Anesthesia and surgical field	EUR 42.25
Position of patient	EUR 18.16
Surgery (prosthetic implant excluded)	EUR 13.01
Suture and dressing	EUR 15.16
TOTAL COST	EUR 97.02

**Table 3 jcm-11-06928-t003:** The total cost of prosthesis refers to the specific components used.

Purchase Prices of Prostheses	
Cup	EUR 786.60
2. Insert	EUR 425.60
3. Head	EUR 280.25
4. Stem	EUR 1420.25
Total without VAT	EUR 2912.70
VAT	EUR 4%
TOTAL COST	EUR 3029.21

**Table 4 jcm-11-06928-t004:** The sum of all cost items calculated by the TDABC approach.

Inpatient Total Cost of Tha	
1. Pre-operative tests	EUR 90.29
2. Hospitalization	EUR 2493.80
3. Operating room	EUR 135.00
4. Consumables	EUR 97.02
5. Prosthesis	EUR 3029.21
6. Personnel	EUR 201.74
FINAL COST	EUR 6002.06

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
