# Peer review of "What Is the Inpatient Cost of Hip Replacement? A Time-Driven Activity Based Costing Pilot Study in an Italian Public Hospital"

_jcm, 2022, doi:10.3390/jcm11236928_

Round 1

Reviewer 1 Report

Dear Authors,

The article makes an interesting contribution to the field on inpatient costs of hip replacement using A Time-Driven Activity-Based Costing approach. However, the article requires significant changes in the structure of the text. It needs to be put in order, and requires some additions. I have comments before the manuscript could be considered for further proceeding and publication:

Abstract

1.      The abstract is not clear, not concise, and requires correction in line with the adopted structure for the abstract of the article, for instance: (1) Introduction, (2) Aim of the study, (3) Materials and Methods, (4) Results, and (5) Conclusions.

Introduction

2.      The sentence in lines 46-47 needs to be corrected from „increasing at rate of 2.7 per year on average....” to „increasing by 2.7% per year on average....”. In addition, for the paragraph on lines 46-53 an explanation is needed why in Italy is increasing total number of hip arthroplasties or what are the tendencies in other countries?

3.      Some paragraphs of the article require redrafting by arranging information relating to Time-Driven Activity-Based Costing. It means that authors should move information from the section Discussion to Introduction (lines 190-193 and 209-219).

There is also a need for a literature review and an extension of the research done so far on the Time-Driven Activity-Based Costing.

Aim of the study

4.      The aim of the study should be identical in the Abstract (lines 25-26) and in the main text (lines 88-91).

Materials and Methods

5.      In the beginning of this section, there should be a separate paragraph “2.1. Study design”, for which a detailed description is needed. In this section it is important to include type of the study, information on the hospital and its characteristics (the structure/organization of health care, the population size being subjected to the health care services and resources, e.g., number of beds, number of medical staff employed there). There is no information, when the study was performed - please indicate the period and the dates. On the end of this section, the authors have to add the reference number of the approval of the Bioethics Committee.

6.      In 2.2. Study participants” must be information on the characteristics of the study group, and criteria of exclusion from the study.

7.      In the next paragraph, „2.3. Measurement” the authors need to explain in more detail the Interviews performed to healthcare professionals” and „Process maps”.

8.      The last paragraph „2.4. Costs analysis of hip arthroplasty” should include a detailed description of individual cost components and a description of total costs; possible mathematical formulas can be presented.

Results

9.      The results need to be given in a more understandable way. The Results require editing and supplementing the description to Figure 1 and Table 1, 2, 3, 4.

10.  In addition, the titles of Figure 1 and Table 2 should be shortened or renamed, so would be more understandable to the readers.

11.  In the Table 3 is „Tot”, but should be „Total”.

12.   Lines 122 and 165-174 contain information that should be included in the Material and Methods section, so it should be moved there.

Discussion

13.  The first paragraph in the Discussion should contain a few concluding sentences describing the „main findings”.

14.  It is also important to compare your study with the studies of other authors - are there any other studies supporting your main result?

Conclusions

15.  The Conclusions of the work should be more informative and response to the purpose of the study. Please correct them.

I would also advise editorial corrections and language improvement to make it more concise.

Please, highlight the changes to the revised version using a different color.

Reviewer 2 Report

This study involved 90 patients undergoing total hip replacement during one year interval with an aim to analyze the medical cost by the TDABC method. As the authors said, accurate understanding of costs is important for the effective implementation of cost saving strategies. Especially, total hip replacement already becomes a standardized procedure for DRG-based payment.

The authors made a plan to collect the relevant data for analyses as following:

1. Inclusion criteria: patients 60–80 years old, suffering from primary hip arthritis with indication for THA. Major comorbidities and trauma are excluded. Patients with intraoperative, surgical or systemic complications were excluded.

2. Interviews performed to healthcare professionals and their observation identified the main activities required for total hip replacement surgery and those involved.

3. Process maps have been created, containing all the actions taken and all the consumables required at each stage of the care process.

4. The amount of time required for each action listed in the process map has also been identified.

5. The cost of the majority of consumer goods used throughout the entire therapeutic process was provided by the Director of hospital's pharmacy by filling out a pre-set table based on the information needs derived from the process maps; the cost of the prosthetic device was determined from the analysis of Regional tenders.

6. Comparing the average monthly pay of the operators with the actual amount of time spent delivering the health service, the capacity cost rate meant as practical capacity of each active operator, was determined. This method was used for all the operators present, taking into account the different remuneration.

7. Finally, we proceeded to the sum of all the calculated costs, thus defining the total cost of hip arthroplasty, from the admission to the patient's discharge.

I noticed several conditions requiring an addressing by the authors.

This study excluded patients with either major co-morbidities/trauma or intraoperative surgical/systemic complications. However, no definition of major co-morbidities and complications were provided.

2. Observation of the healthcare professionals in conducting the main activities required for total hip replacement surgery needs to clarify the basis of estimating the cost of each step.

3. Determining the cost of each working hour provided by the healthcare professionals requires more details to achieve a reasonable estimate. For example, an orthopaedic surgeon usually did many kinds of operation except THR. The cost of each working hour should be differentiated among different surgical procedures that did not get same payment for each working hour.

Since TDABC was adopted to analyze the cost of THR in order to save the expense, precise data is mandatory to convince the readers.   

By the way, DRG-based payment of THR usually contained expense caused 1 or 2 weeks within the hospitalization or surgical procedure. The authors should address the boundary of expense calculation.

I expect to read a proper revision.

Round 2

Reviewer 1 Report

Dear Authors,

Thank you for responding to my previous comments. I believe that the manuscript has improved considerably now.

However, in my opinion, there is one more issue to be clarified - the DRG system in hospital treatment. This abbreviation appears in the abstract, but is not explained in the methodology, and needs to be explained.

In addition, I noticed a mistake on line 230 where it is 98.02 and in the table is 97.02. Please, correct that.

In Table 3, the value of 2.912,70 should be named "Total without VAT".

It would be good to present the cost values in the same manner in all tables, i.e. with two decimal places.

These seem like small details, but really affect people's willingness to read and understand your work and their opinion of its quality.

Good luck with the next step of the processing of your paper.

Author Response

Reviewer 1 point-by-point response

Dear Reviewer 1,

It has been a pleasure to deal with you and an honor to satisfy your considerations.

We explained "DRG" also in the main test, corrected the typo on line 230 (97,02€), and added "Total without VAT" in table 3.

Moreover, the cost values are now presented with two decimal places in all the tables of the main test.

Thank you for your work and for your time

Reviewer 2 Report

As you said, TDABC is a cost study methodology different from the traditional one and different from DRG-based reimbursement.…., that is the reason why I expressed “The cost of each working hour should be differentiated among different surgical procedures that did not get same payment for each working hour.

However, I notice your effort to improve the quality of this manuscript. My questioning was mostly addressed even though the data was not sufficient to conduct a high-quality analysis regarding TDABC .

Author Response

Reviewer 2 point-by-point response

Dear Reviewer 2,

Thank you for your thoughtful observation. You are completely right and different surgical procedures can’t be charged by a similar amount of staff cost. 

Nevertheless, in the Italian context, surgeons (and generally clinical staff) who act exclusively in the Public Health System (that covers health needs of the population as a whole, irrespective people contribution to the fiscal system or their stipulation for a private personal health insurance) are paid by hourly rates unrelated to the services provided. Hourly rates for surgeons’ salaries only depends on fixed parameters related to the experience of the physician as the seniority pay increases. This way, a specific orthopaedic surgeon (Doctor X) is payed the same hourly amount irrespective if he/she perform a hip-replacement or a hallux valgus correction. As a consequence, in Italian public health system, also the operating room activities are not usually accounted on the base of complexity of the procedure performed. 

In this context, exactly following your stream of reasoning, we want to apply Time Driven ABC to evaluate surgical tasks and operating phases that, being more complex, requests for more time to be performed and consequently are more expensive for the public health system. 

This way, given the performing of a hip-replacement will request more time than performing a hallux valgus correction, total staff costs for a hip-replacement is expected to be higher than those for a hallux valgus correction.

Hope this explanation is enough to answer your point; anyway, we would be more than pleased to implement any other recommendations you may have to strengthen our analysis.

We are truly grateful to you for your work and your time.